# The burden of zoonoses in Paraguay: A systematic review

**Liz Paola Noguera Zayas**[1,2]*, **Simon Rüegg**[1], **Paul Torgerson**[1]

**1** Section of Epidemiology, Vetsuisse Faculty, University of Zürich, Zürich, Switzerland, **2** Epidemiology and Biostatistics, Life Science Zürich Graduate School, University of Zürich, Zürich, Switzerland

* lizpaola.noguerazayas@uzh.ch

## Abstract

### Introduction

Underestimation of zoonoses is exacerbated in low and middle-income countries due mainly to inequalities with serious consequences in healthcare. This is difficult to gauge and reduce the impact of those diseases. Our study focuses on Paraguay, where the livestock industry is one of the major components of the country's economy. Therefore, the rationale of this study was to develop a case study in Paraguay to estimate the dual impact of zoonotic diseases on both the human health and animal health sector and thus determine the societal burden of such diseases.

### Methodology/Principal findings

We conducted a systemic review (including a meta-analysis) to assess the burden of zoonoses in Paraguay, including official reports and grey literature of disease incidence and prevalence. We estimated the Disability Adjusted Life Years (DALYs) and Zoonosis Disability Adjusted Life Years (zDALYs) to measure the difference between the current health status and the desired health situation of animals and the Paraguayan population based on 50 zoonotic diseases suggested by the WHO (World Health Organization), OIE (World Organization for Animal Health) and the National Health in Paraguay. The total DALYs represent 19,384 (95% CI: from 15,805 to 29,733), and zDALYs, 62,178 (95% CI: from 48,696 to 77,188). According to the results, the priority pathogens for DALYs are *E. coli*, *Trypanosoma cruzi*, *Leishmania spp*, and *Toxoplasma gondii*. When we include the additional animal health burden, the most important pathogens are *Brucella spp*, *E. coli*, *Trypanosoma cruzi*, and *Fasciola hepatica* for zDALYs.

### Conclusion/Significance

This is the first study to integrate DALYs and zDALYs with important clues related to the health status of Paraguay. Through DALYs and zDALYs, our perspective becomes more complete because we consider not only human health but also animal health. This is important for setting priorities in disease control, especially in a society where livestock contribute significantly to the economy and to human well-being.

**Data Availability Statement:** All relevant data are within the manuscript and its Supporting Information files.

**Funding:** LPNZ received research funding from BECAL ("Becas Don Carlos Antonio López").

BECAL (https://www.becal.gov.py/) funded this study partially as a part of the Paraguayan National Development Plan 2030. The funders had no role in study design, data collection, analysis, decision to publish, or manuscript preparation.

**Competing interests:** The authors have declared that no competing interests exist.

## Author summary

Zoonotic diseases in man are more likely to occur where there is a close association between man and animals. The control of zoonotic diseases requires a "One Health" approach to reduce the risk of such transmission. This represents a challenge in low and middle-income countries, due to inequalities and limited resources, especially in healthcare. For that reason, we need to quantify the impact of zoonoses in those countries such as Paraguay, a major exporter of beef and agricultural products, so that disease control priorities can be set. Through a systematic review, we estimate the Disability Adjusted Life Years (DALYs) and the Zoonosis Disability Adjusted Life Years (zDALYs) based on incidence and prevalence of zoonoses to find gaps between the real and the desired health status of both animals and humans in Paraguay. The zDALY ensures the total societal burden of disease, rather than just the direct human disease burden. We have through zDALYs a more equitable method for disease burden analysis that has a dual impact on human and animal health. Accordingly, we have found that those diseases with the highest zDALY are brucellosis, colibacillosis and Chagas disease.

## Introduction

It has been suggested that 61% of human infectious diseases are zoonoses [1]. Transmission pathways could be by direct or indirect contact, including air, food, and vectors. Presently, disease transmission is exacerbated by anthropogenic activities and global changes (climate change, globalization), causing a high socioeconomic impact [2]. Moreover, low and middle-income countries suffer from major inequalities, especially in healthcare, making people in poverty more vulnerable to easily preventable diseases with hygiene and good food quality.

The Republic of Paraguay is a landlocked, subtropical, and developing country in South America, where the main economic activities are agriculture and livestock production. The country is divided into the *Occidental (Western)* or *Chaco region* (with three departments) and the Oriental (Eastern) region (with 14 departments). Paraguay's population is estimated at 7 million (in 406,752 km$^2$) with 112,000 births in 2018. In the same period, the Ministry of Public Health and Social Welfare registered 31,000 deaths, but the General Directorate of Statistics, Surveys and Censuses estimated 40,000 deaths per year [3]. According to the World Bank, Paraguay has increased its population life expectancy from 2000 (being 70,5 years) to 2018 (74 years old for both sexes).

In 2017, livestock production represented 12.1% of the GDP, and consequently, animal diseases, including zoonoses, represent substantial potential losses to the country's economy [4]. The main animal production corresponded to poultry (24 million from January to April), cattle (13.5 million registered), pigs (0.4 million), sheep (0.3 million), horses (0.2 million) and goats (0.1 million) in 2018 [5]. In 2017, Paraguay exported to more than 140 countries. With such a large livestock sector, the contact between humans and animals results in a higher risk of the transmission of zoonoses. In Paraguay, further (wild) animal-human contact occurs due to deforestation, unplanned urbanization, population growth, invasion of wild habitats, bushmeat, hunting, loss of biodiversity, introduction of exotic (alien) species, certain natural phenomena such as flooding, unequal resource distribution, lack of adequate basic hygiene measures, lack of infrastructure and planning. All these factors provide additional risks for the transmission of zoonotic diseases. Because of this high risk for diseases at the human-animal interface, we hypothesize that zoonoses cause a substantial proportion of the burden of

infectious diseases, this being 149,196 DALYs in 2019 [6]. Our study aims to estimate the burden of zoonoses using disability-adjusted life years (DALYs), as well as an adjusted indicator, zoonotic DALY (zDALY), to include animal loss equivalents in the estimate [7].

These indicators can help prioritize health interventions from a more holistic perspective. They may reveal some opportunities to improve health in Paraguay by harnessing synergies between human and animal health based on a "well-being economy."

## Methods

We identified 50 zoonotic diseases in Paraguay based on the lists of the World Health Organization (WHO), the World Organization for Animal Health (OIE). We also included a list of notifiable zoonoses issued by the government of Paraguay, and other diseases with zoonotic potential that exist in Paraguay and/or South America but which are not considered a priority by the Paraguayan government (S1 Table). As Paraguay is a highly productive country, the priorities related to animal health are mainly of economic interest.

We have considered the lists of all these organizations because they are slightly different, since their objectives vary in relation to their priorities and their concepts (e.g., vectors, reservoirs, target, etc.). By combining these lists, we searched for a more comprehensive view on the influence of zoonoses on several areas, such as human and animal health, socio-economic impact, national and international importance.

### Systematic review

We followed the guidelines for "Preferred Reporting Items for Systematic reviews and Meta-Analyses (PRISMA)" (S1 Prisma checklist) to conduct the systematic review, as depicted in Fig 1 [8,9]. We collected data on incidences and prevalences in Paraguay between January 2000 and December 2019. The information was gathered through five global databases (Web of Science, Scopus, Ebscohost, Pubmed, and Google Scholar) and two Paraguayan databases (Scielo Paraguay, and theses shared by the library of Veterinary Sciences and the Central Library of the National University of Asunción) accessible via websites (details in the S2 Table).

We used Boolean search syntax (OR, AND) to join the scientific nomenclature and popular name(s) of each zoonosis in English and Spanish on the five global databases. For example, **Paraguay AND** *("scientific name of the causative agent"* **OR** *"common disease name in Spanish"* **OR** *"common disease name in English"* **OR** *"an alternative name if needed")*. In the local databases, we only used the scientific name of the zoonotic agent with the disease name in Spanish. We specified the terms and the syntax construction for each search engine in the S1 Text. We complemented the resulting information on these databases to reports published on the websites of the WHO, PAHO, WAHIS, FAOSTAT, CDC, Global Health Data Exchange, the Health Sciences Research Institute in Paraguay, the Ministry of Public Health and Social Welfare and the National Service of Quality and Animal Health (SENACSA). The corresponding links are listed in S2 Table.

We included articles considering the title and the abstracts with the following criteria: reports from Paraguay, any Paraguayan department, city, area, or hospital. We also selected articles of zoonosis with the following terms: "triple border", "South America", "Latin America", "southern cone", "tropical country" (although Paraguay is a subtropical country, it is often confused with a tropical country), "subtropical region", "Neotropical", "the Americas", the "New World", "worldwide", "western hemisphere", diseases in developing countries and diseases in low-and middle-income countries. We also considered reviews and overviews of zoonoses and the presentation of symptoms of undetermined diseases (e.g., pneumonia, kidney injury or parasitic infections).

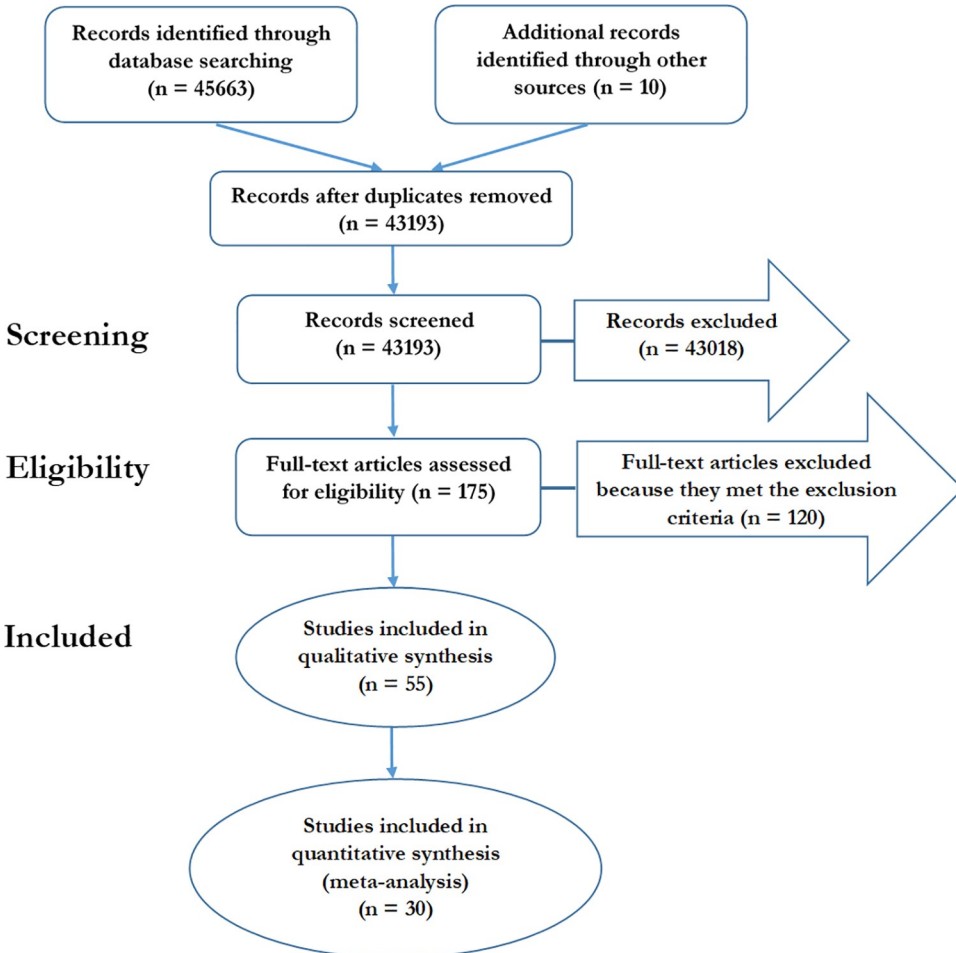

**Fig 1. Flow diagram of the results of the systematic review for the burden of zoonoses in Paraguay, adapted from PRISMA guidelines.**

We excluded articles meeting the following criteria: case studies of frequent diseases when we had enough data, zoonoses occurring outside of Paraguay, Paraguayan emigrants who presented zoonosis cases abroad, laboratory studies without incidence and prevalence, mathematical modelling, questionnaires related to the level of knowledge about certain diseases and information out of the specified period. After article screening, first by title and then by abstracts, we read the eligible papers' full text. Finally, we included the studies in this systematic review with a critical appraisal to estimate DALYs and zDALYs [8,10].

## DALY and zDALY calculation

To assess the burden of zoonoses in Paraguay, we used the DALY incidence-based approach.

$$DALY = YLL + YLD$$

The DALY's burden consists of the sum of *Years of Life Lost* (YLL) due to early death in the population and *Years Lost due to Disability* (YLD) for those who are sick or have sequelae of a certain disease. We estimated the YLL based on the product of the number of deaths (N) multiplied and residual of the life expectancy (L) of the population in Paraguay.

$$YLL = N \times L$$

To obtain the YLD, we multiplied the number of cases (I), the duration of the disease with sequelae zoonosis (L), and the disability weight (DW) representing the disease severity from 0 to 1 (where zero = perfect health, and one = death) [11].

$$YLD = I \times L \times DW.$$

We obtained the residual life expectancy information from the WHO Standard Life Table for Years of Life Lost 2000–2016 (WHO 2018) [12]. When we had no data to estimate DALY, we filled these gaps with global information found on literature and official websites (WHO, CDC, OIE, FAO), such as fatality rate, sequelae, and the disease duration—the URLs are specified in the S2 Table. To calculate the duration of each zoonosis, we considered the total average of the incubation period, the treatment, time to recover, and the sequela (if needed). Regarding tuberculosis, since the reports in human cases did not determine if they were due to *Mycobacterium tuberculosis* or *Mycobacterium bovis*, we used the estimates for the proportion of human tuberculosis due *to M. bovis* infections in humans as a median of 0.3% (range 0%– 33.9%) in the Americas [13]. With respect to zoonotic diarrheal diseases, we applied the methodology of the Foodborne Disease Burden Epidemiology Reference Group (FERG), consisting of a proportion of diarrhea cases due to various etiologies (colibacillosis, salmonellosis, campylobacteriosis), combining with diarrhea data from the Global Burden of Disease (GBD) estimations [14–18]. For cysticercosis, the FERG methodology assumes that 29% of epilepsy cases in at-risk populations will be caused by cysticercosis and the population at higher risk has no access to improved sanitation in countries where pigs are reared. According to the latest data for Paraguay, 11% of the population do not have access to improved sanitation [14,19]. To calculate the cysticercosis burden, we used the DALYs reported for epilepsy for Paraguay on the Institute for Health Metrics and Evaluation (IHME) website [16]. According to its estimates, the epilepsy burden for Paraguay is 12,163 (CI 5,917–21,760).

The zDALY consists of the sum of the DALYs and the animal loss equivalents (ALEs) and represents the burden of disease in society due to zoonoses. This is calculated as [7]:

$$zDALY = YLL + YLD + ALE$$

ALEs are the equivalent impact as the DALY on animal owners. This is estimated through time trade-offs (i.e., the time taken to earn money to replace the animal loss and hence converted to a DALY equivalent). Thus, DALYs and ALEs are integrated. To estimate the ALEs, we considered the annual monetary impact of the animal loss in Paraguay divided by its gross national per capita income (GNI) in US$ [20]. We estimated the average animal loss value per category in Paraguay based on information provided by the website of the SENACSA [21]. Regarding the economic value of cattle, we obtained the prices in US$ from the Rural Association of Paraguay website [22]. For pets (dogs and cats), we estimate the average of the purebred pets' prices, from zero to the mean value based on expert advice who manage market prices in Paraguay and Paraguayan pet selling websites on the internet and social media (further details in the S1 and S2 Tables, and S1 Text). Then, we converted Paraguayan Guarani to US$ at the most recent value at the time of the study [23].

We estimated the 95% probability to account for uncertainty using Monte Carlo analysis. We applied Beta, Gamma, Dirichlet, and Binomial distributions according to the data available. For example, we used beta distributions for the proportions such as incidence and prevalence of diseases; gamma distributions for right-skewed data with a small sample. We used Dirichlet distribution for multinomial or categorical observations, such as leishmaniasis being its main forms: visceral, cutaneous, and mucocutaneous. We performed binomial distributions when the number of observations was fixed, e.g., when researchers selected a determined

number of patients to test for a disease. The distributions and employed data parameters are reported in S3 and S4 Tables.

For each zoonotic disease, we sampled 10,000 iterations from the probability distribution of the parameters, and these samples were summed to give us values to estimate the DALY and zDALY for each zoonotic disease. The median and 95% confidence intervals of DALY and zDALY are reported as the 50, 2.5 and 97.5 percentiles of the 10,000 estimates of the DALY and zDALY. We performed the calculations in R version 3.6.0 using the packages MCMCpack, Prevalence, epiR and rjags.

For the burden of disease estimates, initially, most sensitivity analyses were undertaken for age-weighting and discount rate values, but it is not relevant since these are no longer used. Mathers et al. undertook a sensitivity analysis which essentially ranked the risk factors according to the % of the burden to see how much the total burden will be affected by the uncertainty in estimating any one risk factor [24]. We examined the relative contribution to the total burden by each disease and the relative contributions of the human health effects and the animal health equivalents using the method of Mathers et al. 2006 [24]. Thus, can be seen which diseases and sectors (human or animal health) have the greatest influence on the total burden and hence which would be affected by uncertainty in the data.

## Meta-analysis

According to suitable data previously described, we performed a meta-analysis to get an overall effect size based on the prevalence of the following animal diseases: leishmaniasis, rabies, leptospirosis, ehrlichiosis, scabies, and babesiosis. For human diseases, we performed the meta-analysis of leishmaniasis, leptospirosis, and toxoplasmosis. We applied a random-effect model approach to pool the effect sizes in our meta-analysis (24). In this approach, the model parameters are random variables. We used the funnel plot and Duval & Tweedie's trim-and-fill procedure to gauge publication bias (25). We used R packages: "meta", "metafor" and "forestplot" (18, 19).

## Results

### Systematic review

We identified 43,193 citations between 2000 and 2019 and excluded 43,018 by title and abstract screening, according to the exclusion criteria. We fully assessed 175 articles, resulting in 55 eligible manuscripts that were included in the qualitative synthesis (Fig 1). We performed the meta-analysis of disease prevalence based on 30 articles (those with more than two publications). The results are summarised in the S1 File. We also noticed a substantial increase in scientific publications and reports after 2010, illustrated in Fig 2.

### Burden assessment

We estimated the DALYs of 20 zoonotic diseases and the zDALYs of 25 diseases in Paraguay reported in Table 1. Although the PAHO has recognized Paraguay as free from urban Chagas in 2018, we included it in this work since we found several cases reported in recent years, especially on chronic Chagas [25].

The total DALYs represents 18,424 (14,859–28,750) and zDALYs 62,178 (48,696–77,188) (Table 1). The five pathogens causing the highest burden of disease in decreasing order of DALYs constitute *E. coli*, *Trypanosoma cruzi*, *Leishmania spp*, *Toxoplasma gondii* and *Salmonella spp* (Fig 3). They are responsible for 75% of the disease burden. Considering the ALEs,

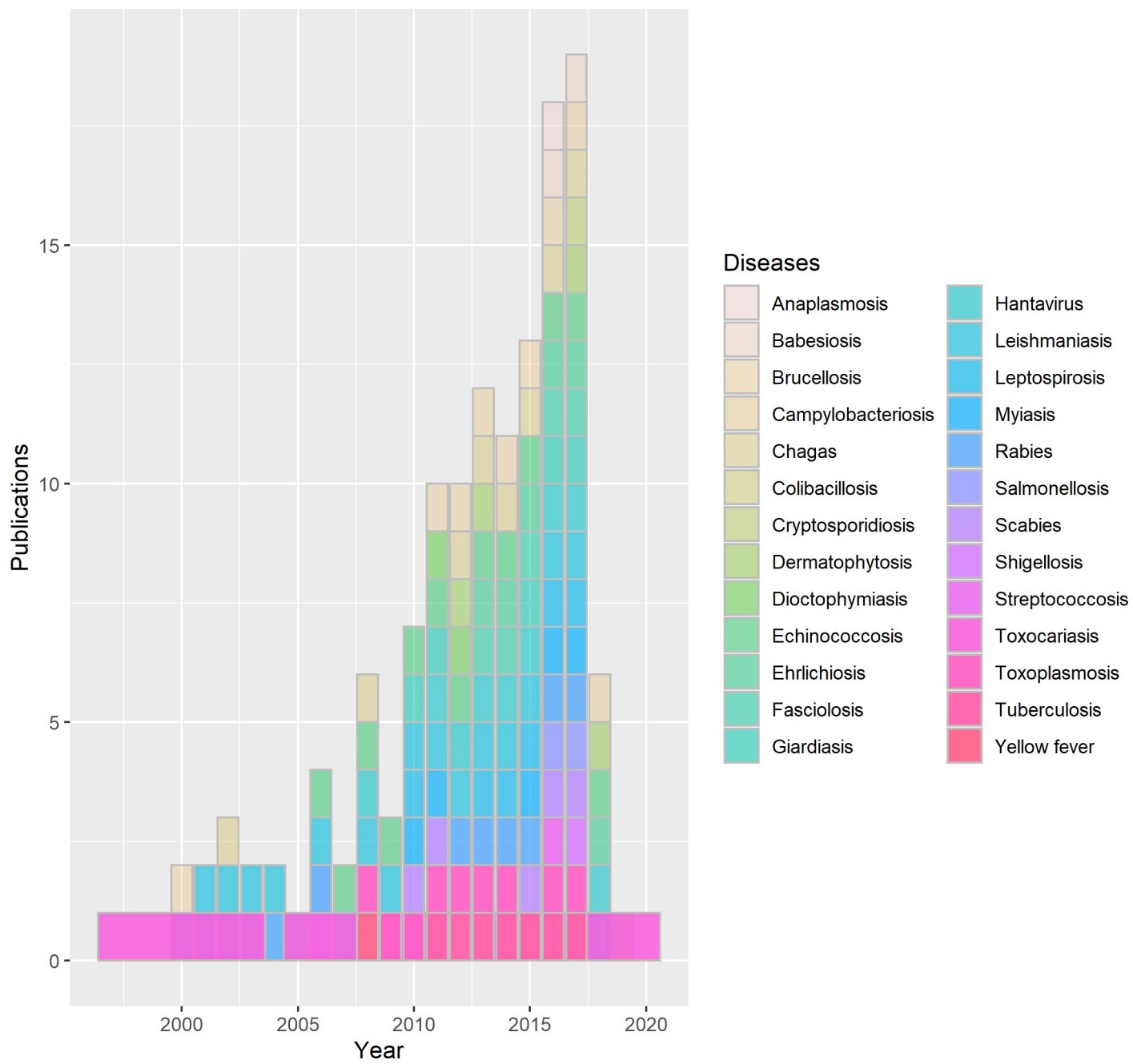

**Fig 2. Diseases publications per year (2000–2019).**

we obtained the following priority (in decreasing order) of pathogens: *Brucella spp*, *Fasciola hepatica*, *Leishmania spp*, *Ehrlichia spp*, and *Leptospira spp* (Table 1).

Regarding zDALYs, it suggests the following priority for zoonoses in Paraguay: *Brucella spp*, *Escherichia coli*, *Trypanosoma cruzi*, *Fasciola hepatica*, and *Leptospira spp* (Fig 4). These high priority diseases would have the greatest influence on the total burden. The burden of disease is also sensitive to the inclusion of ALEs into the total burden estimates (Fig 5).

The number in the right upper corner of each rectangle represents the percentage of the corresponding disease in the total burden of diseases in Paraguay.

**Table 1. Estimated DALYs, YLD and YLL, ALEs and zDALYs of burden of zoonoses in Paraguay with corresponding 95% uncertainty intervals.**

| Disease | DALYs (95% CI) | YLDs (95% CI) | YLLs (95% CI) | Species | ALE (95% CI) | zDALYs (95% CI) |
|---|---|---|---|---|---|---|
| Babesiosis | | | | Cattle | 331 (213–450) | 735 (491–991) |
| | | | | Horse | 300 (191–410) | |
| | | | | Dogs | 104 (21–207) | |
| Brucellosis | 245 (49–505) | 48 (6–60) | 198 (0–463) | | | 7,291 (6,930–7,678) |
| | | | | Cattle | 6,978 (6,684–7,275) | |
| | | | | Dogs | 69 (14–124) | |
| Campylobacteriosis | 678 (144–2150) | 64 (13–216) | 614 (131–1,935) | | | 678 (144–2,150) |
| Chagas | 5,386 (5,281–5,587) | 5,339 (5,254–5,428) | 39 (1.1–231) | | | 5,386 (5,281–5,587) |
| Colibacillosis | 5,412 (1,991–15,531) | 42 (13–92) | 5,349 (1,977–15,465) | Cattle (meat) | 947 (0–2,209) | 6,343 (2,826–16,651) |
| Cryptosporidiosis | 688 (369–1259) | 7 (4–13) | 681 (365–1,248) | Sheep | 175 (120–230) | 865 (543–1,443) |
| Cystic echinococcosis | 375 (249–541) | 181 (120–259) | 194 (128–281) | | | 375 (249–541) |
| Cysticercosis | 388 (188–694) | | | | | 388 (188–694) |
| Dermatophytosis (*M. canis*) | 3 (2–3.4) | 3 (2–3.4) | 0 | Cats | 22 (18–27) | 25 (21–30) |
| Dioctophymosis | | | | Dogs | 95 (69–128) | 95 (69–128) |
| Ehrlichiosis | 132 (0.2–331) | 0.3 (0.1–0.6) | 132 (0–330) | Dogs | 437 (417–458) | 565 (427–775) |
| Fascioliasis | 0 | 0 | 0 | Cattle | 2970 (1,909–4,030) | 3576 (2,512–4,639) |
| | | | | Sheep | 606 (600–612) | |
| Giardiasis | 35 (34–36) | 35 (34–36) | 0 | | | 35 (34–36) |
| Hantavirus pulmonary syndrome | 597 (278–933) | 70 (54–88) | 529 (198–859) | | | 597 (278–933) |
| Leishmaniasis | 2,129 (1,992–2,386) | 2,014 (1,929–2,101) | 110 (16–364) | Dogs | 445 (438–452) | 2,576 (2,438–2,831) |
| Leptospirosis | 671 (338–1,068) | 10 (7–13) | 661 (330–1057) | | | 2,702 (2,293–3,150) |
| | | | | Cattle | 1,743 (1,567–1,914) | |
| | | | | Horse | 292 (215–370) | |
| Myiasis | 9 (8.7–62) | 9 (8.7–9.33) | $6.4e^{-03}$ ($6.3e^{-19}$–54) | | | 9 (8.7–62) |
| Rabies | 148 (114–189) | 9 (8–10) | 139 (105–180) | | | 159 (125–199) |
| | | | | Cattle | 10 (8–13) | |
| | | | | Dogs | 0.8 (0.7–0.9) | |
| Salmonellosis (non–typhoidal) | 949 (550–1475) | 21 (18–26) | 925 (529–1454) | | | 949 (550–1475) |
| Scabies | 228 (226–230) | 228 (226–230) | 0 | Dogs | 6 (2–10) | 234 (230–239) |
| Staphylococcosis | | | | Cattle | 6 (5–8) | 6 (5–8) |
| Streptococcosis | | | | Cattle | 3 (2–6) | 3 (2–6) |
| Toxocariasis | 15 (13–17) | 15 (13–16) | 0 | | | 80 (76–85) |
| | | | | Dogs | 39 (37–41) | |
| | | | | Cats | 26 (25–28) | |
| Toxoplasmosis | 1,139 (1,003–1,342) | 1,098 (979–1,232) | 28 (0.34–200) | | | 1,139 (100–342) |
| Bovine Tuberculosis | 764 (707–823) | 16 (15.9–17) | 793 (397–1,256) | Cattle | 761 (687–842) | 1,561 (1,162–2,027) |
| Total | **18,424 (14,859–28,750)** | **408 (334–500)** | **10,034 (6,500–20,349)** | | **43,123 (31,501–54,558)** | **62,178 (48,696–77,188)** |

## Meta-analysis

Table 2 shows the overall effect of each disease in the meta-analysis with its respective heterogeneity. Among all diseases in the meta-analysis, leishmaniasis in dogs has larger studies, so smaller standard errors.

Cochran's Q test would be underpowered in cases of only two studies, such as rabies. We complement this extent with P-value to determine statistical heterogeneity.

We found a high heterogeneity ($I^2$) across diseases, except scabies (but it is not significant).

We have no evidence of publication bias according to trim-and-fill procedures. See figures in S1 File.

## Discussion

This work represents the first study to integrate the burden of zoonoses to society through DALYs and zDALYs (DALYs + ALEs), using real data from a country and complemented

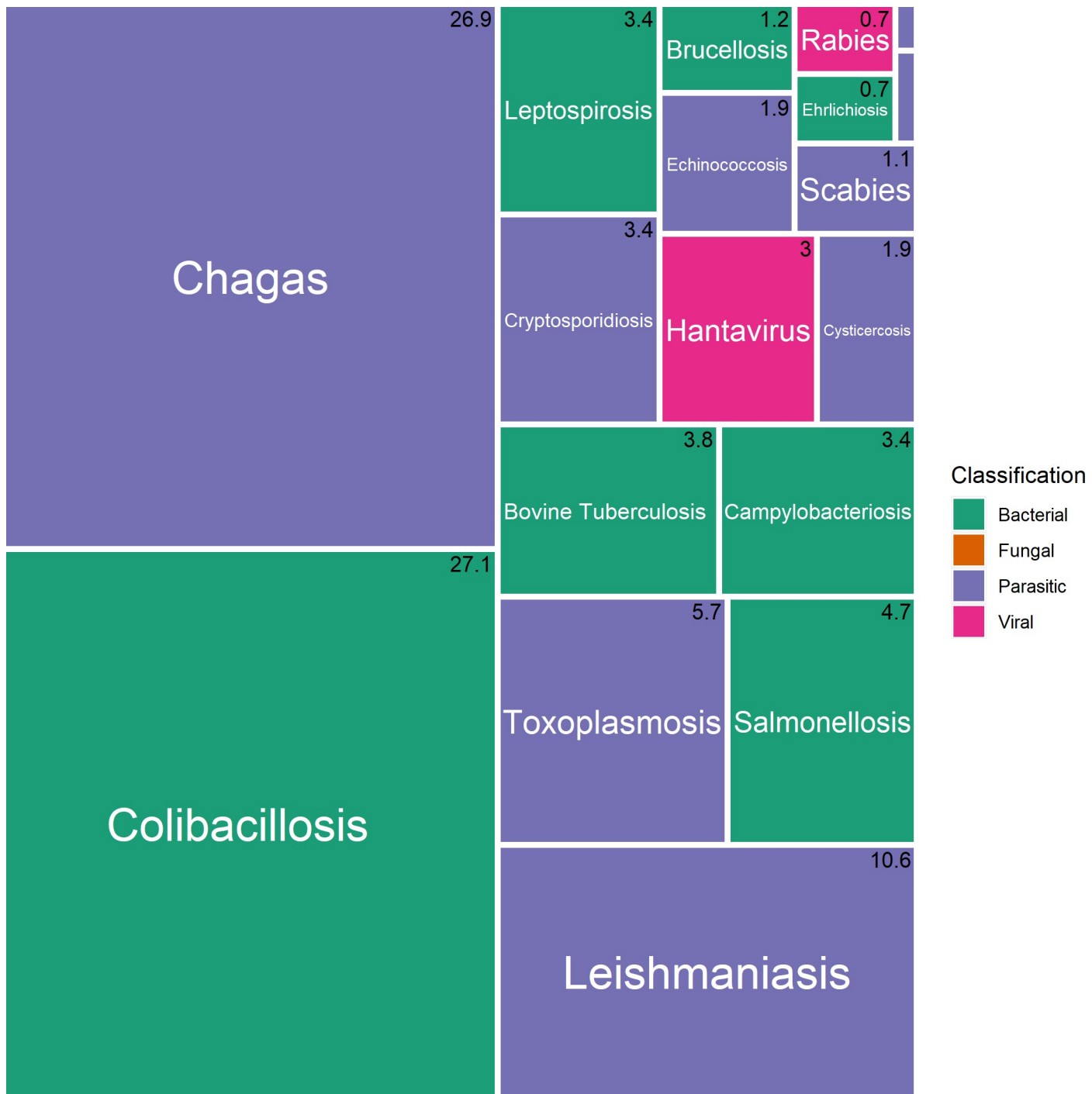

**Fig 3. Relative burden of diseases in Paraguay: DALYs.**

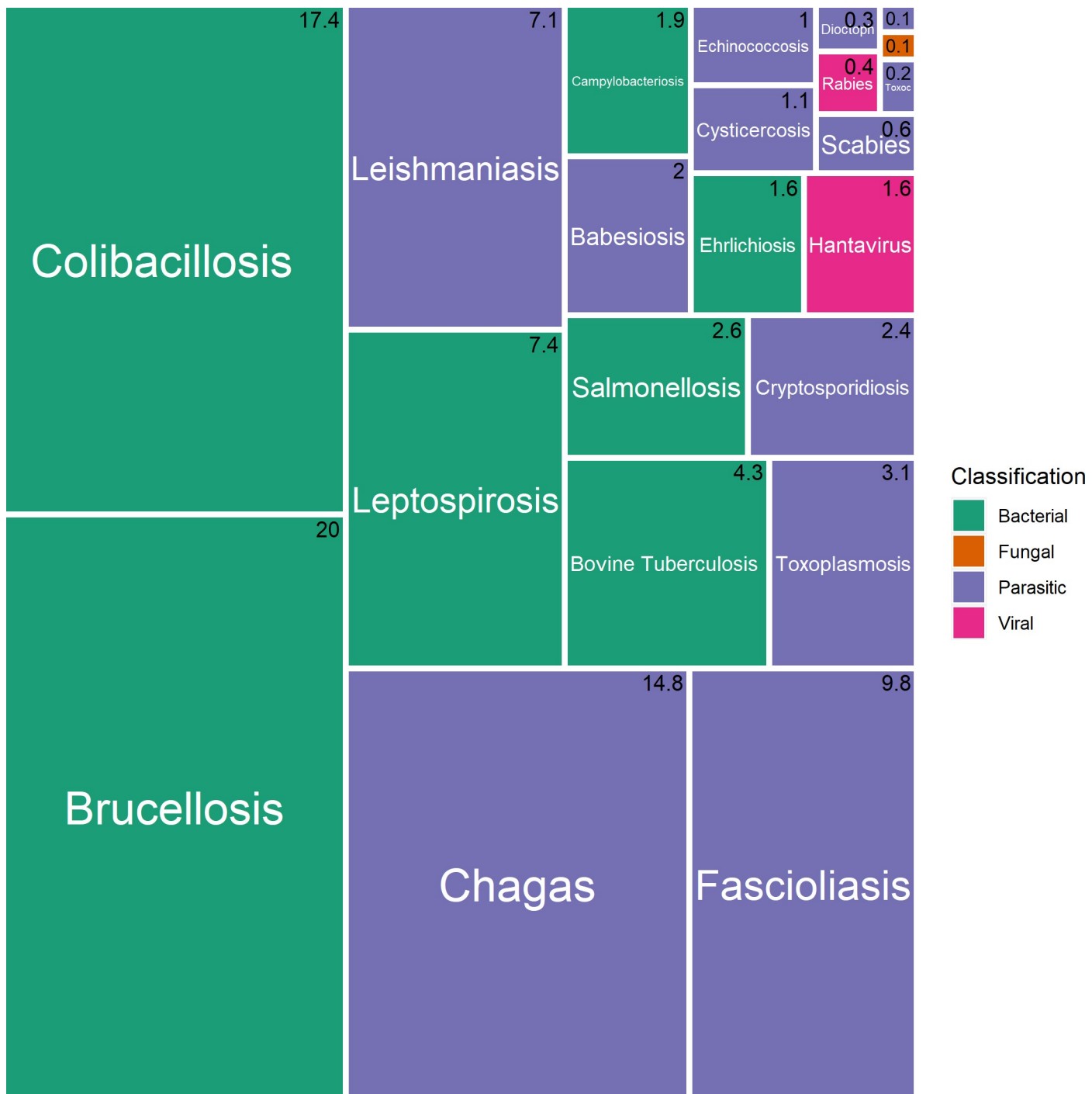

**Fig 4. Relative burden of diseases in Paraguay: zDALYs.**

with conservative assumptions and stochastic modelling. According to the data available and our conservative estimations, zoonoses causing the major burden attributed directly to human diseases in Paraguay were colibacillosis, chagas, leishmaniasis, toxoplasmosis and salmonellosis, whereas the burden attributed to animal health losses (ALE) were brucellosis, fasciolosis, leishmaniasis, ehrlichiosis, and leptospirosis. The total disease burden, combining the burden

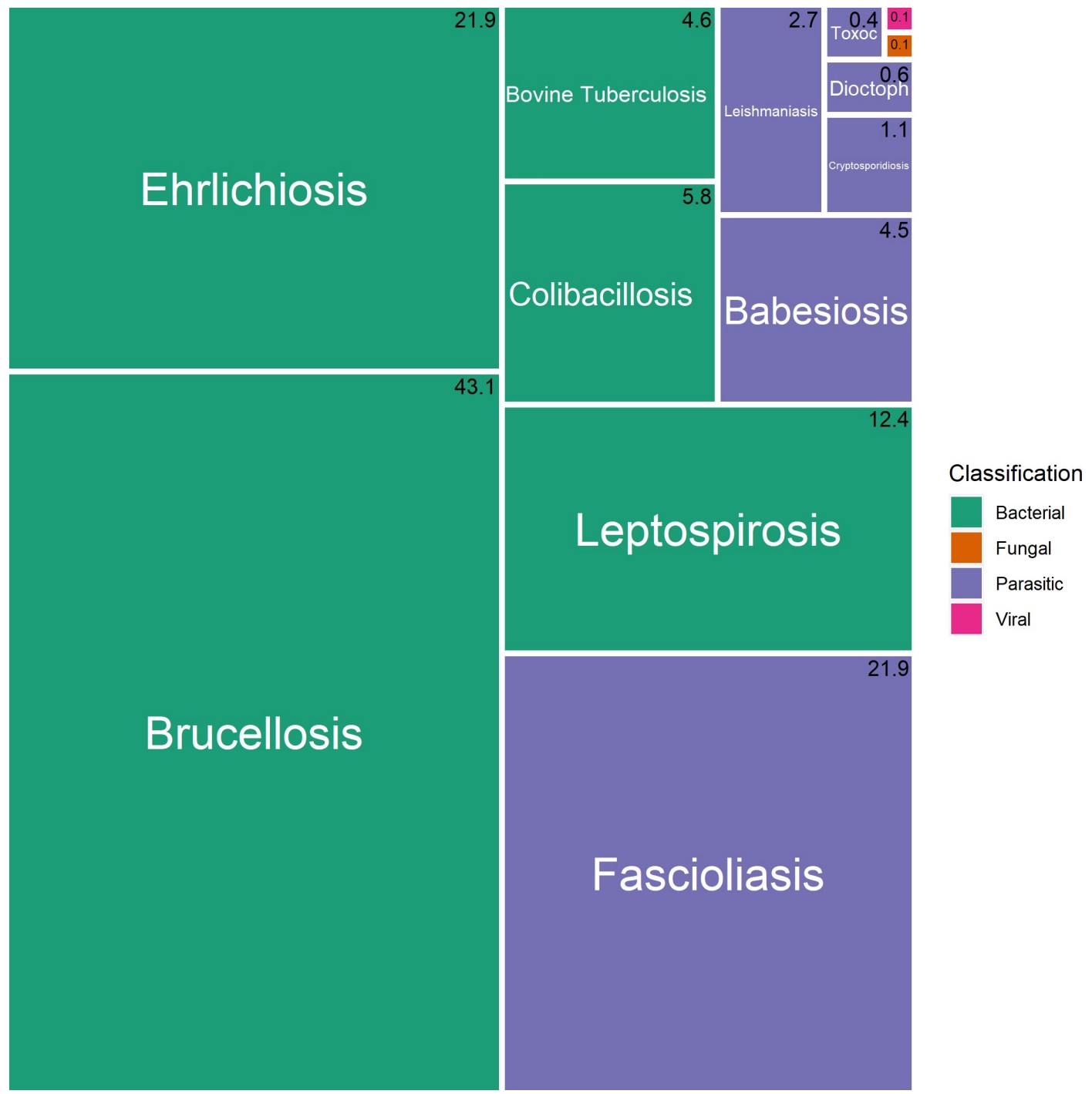

**Fig 5. Relative burden of diseases in Paraguay: ALEs.**

attributed to human and animal burden of zoonotic diseases (zDALYs), has resulted in a change of priorities to brucellosis, colibacillosis, Chagas, fasciolosis and leptospirosis. We observed slightly different results according to the approach, being not opposed but complementary. A high burden of zoonotic diseases in animals means a higher risk of spreading disease to humans if preventive measures are not taken. Our dataset may suffer bias, namely the

**Table 2. Summary of the meta-analysis.**

| Disease | Overall effect (95% range) | Tau$^2$ | I$^2$ | Q | P–value |
|---|---|---|---|---|---|
| Leishmaniasis (animal) | 0.33 (0.18–0.44) | 0.041 (SE = 0.024) | 99.85% | 1788 | < .0001 |
| Leishmaniasis (human) | 0.60 (0.29–0.91) | 0.0740 (SE = 0.0758) | 98.26% | 58 | < .0001 |
| Brucellosis (animal) | 0.03 (0.01–0.05) | 0.0002 (SE = 0.0003) | 99.80% | 502 | < .0001 |
| Rabies (animal) | 0.62 (0.20–1.03) | 0.089 (SE = 0.13) | 98.05% | 51 | < .0001 |
| Leptospirosis (animal) | 0.29 (0.12–0.46) | 0.022 (SE = 0.023) | 98.00% | 80 | < .0001 |
| Leptospirosis (human) | 0.06 (0.02–0.11) | 0.0008 (SE = 0.0016) | 70.46% | 3.9 | 0.066 |
| Ehrlichiosis (animal) | 0.33 (0.09–0.57) | 0.059 (SE = 0.0496) | 99.71% | 1875 | < .0001 |
| Scabies (animal) | 0.11 (0.08–0.14) | 0 (SE = 0.0020) | 0.18% | 1.9 | 0.3943 |
| Babesiosis (animal) | 0.21 (0.05–0.38) | 0.025 (SE = 0.0230) | 91.14% | 21.11 | < .0001 |
| Toxoplasmosis | 0.4 (0.09–0.71) | 0.12 (SE = 0.088) | 99.54% | 1005 | < .0001 |

lack of available data from several geographical areas, diseases, vulnerable human populations, and neglected animal populations. We chose Monte Carlo method for stochastic modelling since it has been widely used to model the probability distribution of diseases due to its simplicity and flexibility. All the outputs consist of estimations based on data that we found through the systematic review.

We assume an underestimation of the total burden of zoonoses in Paraguay due to underreporting, under-ascertainment, underdiagnoses, and misdiagnoses. According to the Ministry of Public Health and Social Welfare, 8,971 of 31,258 deaths (28.7%) occurred without medical assistance in 2018, and similar cases happen every year. It is estimated that approximately 8,598 deaths per year are not registered in this country [3]. The cause of several of those deaths would have a high probability of under-reporting and under-ascertainment. Currently in Paraguay, zoonotic diseases with surveillance program lead by the Ministry of Health are bovine tuberculosis (National Program for Tuberculosis Control–PNCT), rabies (National Program of Zoonotic Diseases and National Rabies Center–CAN), Chagas, and leishmaniasis [26]. Our observation suggests underreporting from rural areas and inland Paraguay since most scientific articles correspond to the capital, Asunción (or metropolitan area), and the Central Department due to infrastructures and resources are mainly in this area. This illustrates the challenge of Paraguay, being a "centralized" country where most services (especially health, sciences, infrastructure, and resources) are in "Gran Asunción" and the Central department, leaving other departments and most rural areas neglected. Among studies outside Central Department, they are primarily contained in theses that are not easily accessible.

Another factor that would increase zoonosis under-ascertainment is illiteracy, in addition to contributing to the spread of zoonoses. Until 2018, 6% of the Paraguayan population remained illiterate [27]. The challenge to prevent zoonoses (especially foodborne diseases) is even more critical with an illiterate population. For instance in 2017, in Caaguazú (the main dairy producer and supplier of the country), 720 workers in 360 dairy herds were interviewed, 11.7% of them were illiterate, 64% stated that they did not know what brucellosis is, and 73.3% had no knowledge of how to prevent brucellosis. All of them also used to consume unpasteurized dairy by-products, mainly Paraguayan cheese [28]. Dairy farm owners, the Ministry of Health and the National Service of Quality and Animal Health (SENACSA) are primarily responsible for educating and sensitizing dairy workers to follow preventive practices. A law (Nr. 5804) established in 2017, imposes a national system to prevent labor risks in public and private entities [29]. Another issue is that workers from informal sectors are difficult to protect, so they are more vulnerable to injustices and diseases. In 2014, Paraguay registered 78% of informality rate being 88.6% in rural areas [30]. Informal sectors and poverty are frequently

     

associated; approximately 5.9% of the Paraguayan population lived in poverty with an income below US$ 3.20 per day in 2018. This sector is the most vulnerable to hunger and diseases in the country [31].

Paraguay is a major food exporter, but malnutrition and hunger are still a concern. The Nutritional Food Surveillance System (SISVAN) in Paraguay reported that 11.4% of children under five years old suffer from chronic malnutrition. According to the FAO, undernourishment has increased over the last three years in this country as well as obesity. Based on the reported data, Paraguay has reduced poverty in the last years, increased food production, but undernourishment has also increased, proving that inequalities are part of this process. Currently, Paraguay is the third country in South America with the most inequalities after Brazil and Colombia [32].

Part of the zoonoses under-ascertainment would be related to demographic characteristics of the current Paraguayan population, which are only based on estimates since the national census carried out in 2012 was not completed as established. According to the international standards, a census should cover at least 90–95% of the population to be valid, but in Paraguay, it was only covered 74.4% of the population [33]. In addition, UNICEF estimated in 2013 that one in four children was not registered in Paraguay [34]. This condition increases our uncertainty of gauging the true impact of zoonoses in Paraguay. In our search, we observed that certain diagnoses are symptomatic (e.g., diarrhea, fever) rather than etiologic resulting in underdiagnosis of zoonoses. For instance, some reports on heart disease do not explain the etiologic cause, such as cases of "dilated cardiomyopathy" can be related to toxoplasmosis, Chagas, histoplasmosis, Lyme disease, rickettsial diseases, nutritional causes, among others [35–37].

Misdiagnoses contribute to the underestimation due to several causes, such as cross-reactivity in diagnostic tests, e.g., serological leishmaniasis test [38,39]. Furthermore, not all people can afford a gold standard test, pay for a test repetition, or spend money on a second medical opinion. In case of leishmaniasis, positive pets are euthanized as a disposition of Ministry of Health in Paraguay. This represents an inconvenience when the definitive diagnosis is not confirmed by a gold standard test. As a result, the risk of facing other pathologies that we ignore is higher, and owners would lose needlessly lives of sick pets caused by other diseases, even non-zoonotic ones. The National Program of Zoonoses Control in Paraguay manages the diagnosis of leishmaniosis without a gold standard test. In 2016, as a routine exam of leishmaniosis in 6,672 dogs, using rK39 test, 1,586 resulted positive of which 1,038 positive dogs were culled without a gold standard test (as data reported in the 6th World Congress on Leishmaniasis). Taking into account that an rK39 test has a specificity of approximately 82% [40]; applying to the previous case, 851 dogs would have been culled unnecessarily by misdiagnosis.

In 2018, the Pan American Health Organization certified Paraguay as a "country free of domiciliary vector transmission of Chagas disease caused by *Trypanosoma cruzi*". Despite this certification, Chagas represents one of the highest DALYs among zoonoses, especially those due to chronic Chagas. Moreover, we also found acute cases of Chagas, but the reports did not specify the transmission form (whether they were urban or sylvatic). Although Paraguay is a "country free of domiciliary vector transmission of Chagas", the work related to this disease has not finished yet, and it must not be neglected. The WHO has estimated that approximately 6 to 7 million people are infected by *T. cruzi* worldwide, being the most affected Latin America. The triatomine bug is the vector that transmitted this disease. Chagas is not only considered a vector-borne disease but also a foodborne, bloodborne and congenital disease. Over centuries, the vector of Chagas or American trypanosomiasis has had an "evolution" or adaption, affecting at the beginning only wild mammals until also spreading to domestic mammals. In the past, Chagas was usual in rural areas but the urban cases have been increasing lately due

mainly to the migration of infected people and anthropogenic activities that cause changes in the environment, increasing the risk of contact with the vector. Chagas or American trypanosomiasis is endemic in Latin America, and since 1990, the WHO has reported a reduction of transmissions. The most affected countries in South America are Brazil (174,194.22 DALYs), followed by Venezuela (27,037.40 DALYs), and Argentina (23,552.58 DALYs); Uruguay being the least affected (350.72 DALYs) based on the estimations of the Global Burden of Diseases [41]. Concerning Chagas in pets, there were a few studies of seroprevalence of Chagas in dogs in Latin America such as Argentina, Brazil, Chile, Colombia, Ecuador, Mexico, Venezuela but none in Paraguay [42–48]. Several other mammals are a potential reservoir of Chagas. This is one of the reasons that eradication of this disease is not feasible, but its impact can be reduced not only by taking care of the animal and human health but also of the environment.

Foodborne zoonoses are the most frequently underreported since they principally cause diarrhea, and the treatments are mainly symptomatic. Paraguay registered 92,466 cases of diarrhea, being 83,140 classified as parasitic origin, 216 as food poisoning, and the remaining 9,110 were unspecified, according to Paraguay statistical yearbook 2017. Since diarrhea cases are frequently underreported, we used the estimates of diarrhea for Paraguay of 10.3 million (9.5 million—11 million) [49]. The most common bacterial zoonotic diseases were colibacillosis, nontyphoidal salmonellosis, and campylobacteriosis [50]. In Paraguay, colibacillosis cases were mainly reported in children and condemned beef in abattoirs through certain theses. Worldwide, Shiga toxin-producing *E. coli* (STEC) causes outbreaks mainly associated with undercooked or raw beef and its products; also, fecal contamination of vegetables and water. The reservoirs of *E. coli* are mainly cattle, goats, sheep, and deer. Other animals can be infected by this pathogen, such as dogs, cats, chicken, horses, pigs, rabbits, and turkeys. Besides being *E. coli*, a foodborne and waterborne disease, it can also be transmitted from person to person (through unwashed hands). The O157:H7 serotype is the main related to STEC and can cause hemolytic uremic syndrome (HUS). Lately, it seems a lack of updated outbreaks of *E. coli* in Latin America, whereas for the USA, the *E. coli* infections and outbreaks are constantly updated by the CDC. Therefore, we easily found more cases in the latter country.

In Paraguay, we only found cases of campylobacteriosis and salmonellosis in children. The WHO affirms that worldwide, campylobacteriosis and salmonellosis are one of the main causes of diarrhea of approximately 1 in 10 people yearly. Both foodborne and waterborne diseases usually cause mild symptoms, but they can generate some complications. In the case of campylobacteriosis, among its complications are bacteraemia, hepatitis, pancreatitis, and abortion. Rare cases manifest post-infection complications, including reactive arthritis for several months, Guillain-Barré syndrome, a polio-like form of paralysis. *Campylobacter spp* can be found in chicken, cattle, pigs, sheep, shellfish, ostriches, cats, and dogs. In the case of salmonellosis, the transmission can also be from person-to-person (fecal-oral route)–besides contaminated food. *Salmonella spp* is in chicken, cattle, pigs, dogs, cats, and wild animals such as birds and reptiles. Most of the cases of salmonellosis are considered sporadic, according to the WHO. *E. coli*, *Campylobacter spp*, and *Salmonella spp* have increased antimicrobial resistance as well as awareness about this issue. WHO and FAO formed an alliance to prevent foodborne diseases creating the International Network of Food Safety Authorities (INFOSAN) and to contribute to early detection and response they created an Advisory Group on Integrated Surveillance of Antimicrobial Resistance (AGISAR). Only at the end of 2018 was banned the sale of antibiotics without a prescription in Paraguay.

Within the parasitic causes of diarrhea, we identified giardiasis primarily in children [51]. Giardia intestinalis, lamblia or duodenalis is distributed worldwide, but with a low report because most infections are asymptomatic. Other parasitic zoonoses found in Paraguay with different symptoms were cystic echinococcosis (*Echinococcus granulosus*)–identified as

surgically treated cases—and toxocariasis (*Toxocara canis*) in children. There are low reports of these diseases in Paraguay, only in five countries of South America (Argentina, Chile, Peru, Uruguay, and Brazil) cystic echinococcosis is considered endemic as stated by PAHO from 2009 to 2014. Paraguay is likely to have more echinococcosis cases than those found, but we have almost no data due to missing official reports or surveillance of this disease. Cases of giardiasis, echinococcosis, and toxocariasis can be prevented with good hygiene practices that are difficult to apply in deprived places without safe drinking water or lack of access to improved sanitation. The presence of stray dogs aggravates the risk in urban settings, and the same problem occurs in rural areas where certain farm animals are free to roam.

We did not find any report of cysticercosis in Paraguay. Although the Institute for Health Metrics and Evaluation (IHME) data estimated for cysticercosis a DALY of 2,531 (1,395–4,002), we consider this figure represents an overestimation. In 2018, only 359,604 pigs were registered in Paraguay, most of them corresponded to intensive farms with a low transmission risk. Based on the Foodborne Disease Burden Epidemiology Reference Group (FERG) methodology [52], we estimated 388 (188–694) DALYs instead, and recommend studies on neurocysticercosis, especially in areas that lack basic sanitation with free-range pigs. The PAHO estimates 14.9 million people with neurocysticercosis and 1.35 million with epilepsy due to this disease in Latin American and the Caribbean.

Another important parasitic foodborne disease among the priorities is fascioliasis or liver fluke (*Fasciola hepatica*) that was only reported in cattle and sheep in Paraguay, and it is the sole foodborne trematode infection registered in this country. We have not found human cases, but the risk is higher since the burden in animals is also high. The WHO affirms that about 2.4 million people are infected worldwide. The PAHO has declared that the most affected places by this disease in South America are the Andean highlands, mainly indigenous communities. Paraguay declared in its census that the estimated indigenous population was 112,848 people in 2012. However, there are no studies or surveillance in these communities about this disease.

With respect to leishmaniasis, it was difficult to identify if the reported cases were part of a coinfection or a primary infection. This information would help assess the extent of the specific disease burden, and how much is attributable to coinfection in patients with HIV or other immunosuppressive diseases [53]. Comorbidities of zoonoses were not common to find, but zoonotic infections such as toxoplasmosis and cryptosporidiosis with HIV co-infections were more frequent. According to the Global Health Data Exchange, the HIV burden of Paraguay was 69,576 (45,151–96,058) in 2017 [41], which is still higher than the estimated total burden of zoonoses in people and animals (zDALYs)—62,178 (48,696–77,188).

Toxoplasmosis in humans represents one of Paraguay's most studied infectious diseases, especially in pregnant women and children (congenital toxoplasmosis, being the most common transmission form). Ocular toxoplasmosis is usually reported and, to a lesser extent, cerebral toxoplasmosis as co-infections mainly with HIV [54,55]. Cats are the primary host of *Toxoplasma gondii*, but it can infect a wide range of wild and domestic animals, including dogs. Toxoplasmosis is one of the diseases that can be mainly prevented with good hygienic practices, especially related to food.

We identified a high human tuberculosis incidence of 36.7 per 100,000 inhabitants in Paraguay [56]. Since zoonotic tuberculosis is mainly due to *Mycobacterium bovis*, we were interested in finding the infections caused by this pathogen. However, reports on human tuberculosis in Paraguay did not specify if they were caused by *M. tuberculosis* or *M. bovis*. For that reason, we used the estimates for the proportion of human tuberculosis due to *M. bovis* as a median of 0.3% (range 0%– 33.9%) in the Americas [13]–since Paraguay does not count for this proportion. Our estimates for bovine tuberculosis in human cases are 764 (707–823)

DALYs and when we include all the cases of human tuberculosis the DALYs sum 25,460 (23,590–27,383) in Paraguay. We only considered the results of bovine tuberculosis because our focus was on zoonotic tuberculosis. In the future, it will be useful a study about tuberculosis as an anthropozoonosis to understand better the dynamic of this disease since there are almost no data globally about it, and this is an important approach in prevention.

Leptospirosis has a high prevalence, predominantly in the most disadvantaged areas and sectors of the country. In Paraguay, its incidence is related to flooding and living conditions of people in poverty (aggravated by the lack of infrastructure). Poverty is also associated with other zoonoses such as scabies, larva migrans, fungal skin diseases, vector-borne diseases (dengue and other arboviruses), and animal-related problems; namely animal bites and contact with poisonous animals [57–59]. All these diseases mentioned above are underreported in Paraguay and low- and middle-income countries, underestimating their actual impact.

Hantaviruses (spreading mainly by rodents) present two main forms: hemorrhagic fever with renal syndrome (HFRS)—reported in Asia and Europe—and hantavirus pulmonary syndrome (HPS) in the Americas. The first registered outbreak of hantaviruses in Paraguay was in 1995 in the Chaco region [60,61]. As a result, most studies of HPS have been conducted in this region. Currently, hantavirus is considered endemic of Paraguay's main ecological regions: the Chaco and the Atlantic Forest [62]. We notice certain fluctuations in hantaviruses case records, such as in 2011, 74 HPS cases, and none in 2012. In 2017, the Paraguayan statistical yearbook reported 10 HPS cases, whereas the WAHIS (World Animal Health Information System-OIE) did not specify the number of cases. We consider that the reporting systems need to be improved. Countries in Latin America such as Argentina, Chile, and Panama have registered increased cases of hantavirus infection in the last years.

With respect to animal diseases, Paraguay has larger amount of data about productive and reproductive animal diseases due to the traceability system of its livestock, playing a fundamental key in the economy of this country. We found more data on tuberculosis in cattle, leptospirosis in livestock, brucellosis in cattle, *E. coli* in beef products, rabies in various species and leishmaniosis in dogs. The animal health service in Paraguay (SENACSA) has registries at national level of rabies, tuberculosis, and brucellosis in cattle. We observed that cattle tested for tuberculosis (< 30,000 per year on average) are less than those tested for brucellosis (100,000 per year), being also another possible source of bias since some diseases are under surveillance more than others. As beef is the main exported meat, this country spends most of its resources for surveillance of cattle diseases and inevitably neglects other important species. In Paraguay, the least studied farm animals but with a considerable number per inhabitants are chicken (the most produced in the country) and pigs (the third largest local production, after cattle)—see Fig 6. In this aspect, it is crucial to adjust lenses and refocus the priorities to control diseases. Although livestock production is indeed a priority because it contributes to the country's economic growth, it can also contribute to poverty if diseases of surrounding species are not controlled and prevented (this factor may also affect the economy directly and indirectly). Another risk consists of bushmeat consumed by a small proportion of the Paraguayan population. Some farmers breed wild animals for consumption without any veterinary control or regulations in the country. Also, activities related to hunting have a higher risk of zoonotic infections due to wild animals are not under food quality control. Therefore, more education and control are needed in this regard.

The zDALYs metric is intended to estimate the burden of zoonoses combining the burden of disease due to human and animal morbidity and mortality. For zDALYs, we consider the time lost in years of life that zoonotic diseases cause through human morbidity and mortality and the time trade-off to "recover" the animal loss due to those diseases. This makes estimates of the animal loss equivalents (ALE) on pets and wildlife challenging, because market values

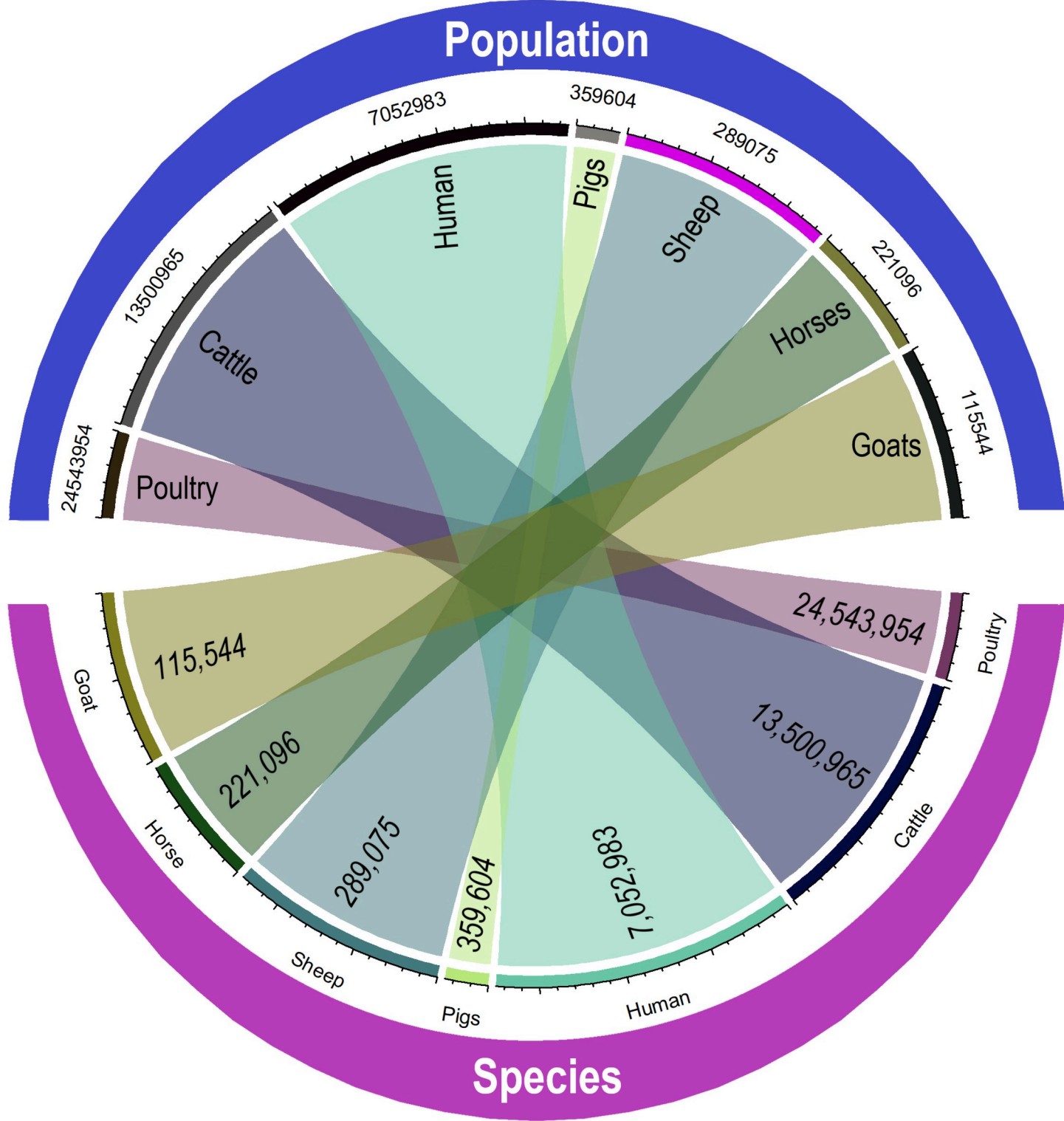

**Fig 6. Diagram of the human and livestock population related to their proportion of publications and studies in Paraguay.**

do not reflect the complex relationship between people and pets or wildlife, and their value in the ecosystem. For a robust estimation of the time trade-off, the moral and emotional factors require further research. This indicates that we may have underestimated the ALEs of zoonoses in pets; and for the moment, we cannot estimate ALEs for wild animals [63–67]. For DALYs, the emotional value is not directly estimated but this is intrinsically included in the disability weight estimations.

We estimated for the first time the DALYs on ehrlichiosis, scabies, and dermatophytosis (*Microsporum canis*). These diseases affect animal and human life quality, and zoonoses such as ehrlichiosis can even be a fatal tick-borne disease if it is not treated. According to our estimations, ehrlichiosis causes the second highest burden in animals (mainly in dogs) but it is not even included among the disease priorities of Paraguay. In this country, the first ehrlichiosis cases reported in scientific studies were from 2013–2014 in dogs, and in the last years some cases in humans. This disease has been reported in most countries of South America.

Certain fungal diseases are considered sapronoses but to some extent its origin is related to animals, such as their feces. Fungal diseases are mainly manifested as coinfections in patients with immunosuppression. We found cases of *Histoplasma capsulatum* (endemic in Paraguay), *Sporothrix schenckii* complex (report of a family case transmitted by a cat), and *Crytococcus neoformans* with insufficient data to estimate DALYs [53,55,68,69]. Although histoplasmosis is considered endemic in South America, only in Brazil, Argentina, and Colombia have published studies about this disease transmitted mainly through bat and bird droppings and associated with HIV- positive patients [70].

Animal bites are not zoonoses but contribute to the burden of diseases through injuries causing zoonotic infections and deaths in cases not treated in time. In 2013, 5799 patients were treated for animal bites (dogs, cats, monkeys, bats and rodents) and in 2015, Paraguay registered 169 cases of snakebites (*Bothrops*, *Crotalus*, and other unidentified species) [71–73]. In addition, snakebites cause significant losses to livestock in some countries. However, there is no evidence of how much livestock farmers lose due to snakebites in Paraguay [74], and how much these agricultural activities damage the environment. To study the impact of animal bites, surveillance is needed, especially for snakebites (venomous animal contact). Although the last case of human rabies was in 2004 and the last rabies cases in dogs were in 2015, Paraguay should be vigilant due to cases of rabies have recently been reported in Argentina. The PAHO reported in 2017 that canine rabies was still endemic in Bolivia (another country bordering Paraguay), Guatemala, Haiti and the Dominican Republic [75]. A study in 2011 in Paraguay, after two confirmed cases of rabies in dogs, demonstrated that 78% of pets have not been vaccinated (49/63) in Loma Plata, Chaco [76]. Unfortunately, unvaccinated animals are common in various parts of the country, especially in peripherical and rural areas, representing a higher risk for rabies outbreaks [77]. Despite no cases in humans, it remains a burden of rabies due to the use of post-exposure prophylaxis (PEP) following animal bites. Although the costs related to rabies are higher and more than the PEP, when we also consider the pre-exposure prophylaxis (PreEP) which varies according to the country. For Paraguay, we did not find official figures that represent that cost. We assume that if we include more costs for rabies prevention, the burden will increase as well as all the zoonotic diseases under surveillance in the country. However, we cannot add any extra amount that is not official since, in low and middle-income countries such as Paraguay, owners are in charge of the rabies vaccination and sterilization of their pets in most cases. Another source of rabies in livestock is wildlife, in particular bats and foxes [78]. Consequently, rabies incidence has increased in cattle [79]. In 2002, a case of human rabies transmitted by a bat was reported in Paraguay [80]. In the process of recent anthropogenic activities, wildlife has fewer natural habitats and Paraguay has lost

biodiversity resulting in an increased risk of zoonoses [81–83]. Paraguay has lost a vast area of its humid primary forest, being decreased by 31% from 2002 to 2020.

We only found a single outbreak of yellow fever in 2008 (after 40 years without reported cases of this disease) in the department of San Pedro, Caaguazú, and Central (San Lorenzo) [84,85]. The same year in San Pedro and Central Departments, researchers tested some primates for yellow fever (Cebus sp: 31) and (*Alouatta caraya*: 13), with negative results [86]. In South America, yellow fever is mainly sylvatic with non-human primates being the principal reservoir. This is in contrast to Africa where it is primarily human-to-human transmission, via vectors [87]. According to the latest report of the PAHO, only Bolivia, Peru, and Brazil registered cases of yellow fever, being the latter country with the highest seasonal cases in 2019 in South America [88].

Another disease present in Paraguay and poorly studied is tungiasis *(Tunga penetrans* and *Tunga trimamillata)*. There is evidence that this disease has been present since pre-hispanic America [89]. However, it is ironically one of the least studied disease in this continent. The most affected population is in extreme poverty and lacks medical assistance [90]. Certain affected people remove *Tunga penetrans* themselves, leading to an underestimation of the incidence. Furthermore, tungiasis does not represent a disease of major concern, but it could lead to more severe infections such as tetanus. In Paraguay, myiasis is a frequent parasitic infestation in animals, especially in warm seasons (the predominant climate in this subtropical country). However, veterinarians usually do not report it. Regarding human cases of myiasis, we found some records, but they did not specify what part of the body was affected, making it difficult to properly estimate the disability weight. As a result, for the DALYs of myiasis estimations, we do not consider any sequelae since it varies considerably according to where the infestation is located. In South America besides Paraguay, a few cases of myiasis were reported in Argentina, Brazil, Equator, Peru, Uruguay, and Venezuela; some of the registered cases were about travellers returning from one of the mentioned countries [91–94]. Other zoonoses that are rarely reported hence of low risk include sparganosis, and dirofilariosis [95,96]. Diphyllobothrioisis has not been reported in Paraguay, but it is present in neighbouring countries [97].

Among zoonotic influenza, Paraguay was also part of the swine influenza (H1N1) pandemic in 2009. Between 2009 and 2010, Paraguay reported 1025 cases and 47 deaths according to the PAHO. The vaccination helped to control this disease as well as a national response plan to an eventual influenza pandemic created in 2003. After 2010, it has not been published reports of outbreaks. Concerning avian influenza (H5N1), Paraguay has no reports as well as any country in South America.

Until now, the notifiable diseases that seem still absent in Paraguay or at least without reports are: Q fever, tularaemia, venezuelan equine encephalitis virus, rift valley fever—all of them are vectorial—and, nipah virus encephalomyelitis (non-vectorial). Concerning the epidemiological information of Q fever *(Coxiella burnetii)*, it has been considered that its distribution is worldwide. However, Q fever is not commonly reported in neighboring countries of Paraguay either, there were only some cases reported in Brazil, and evidence of circulation in Argentina [98–101]. Brazil identified more cases of Q fever when they started including this disease as part of the differential diagnoses list for flu-like diseases such as dengue [102]. Tularaemia has not been reported in South America thus far. Venezuelan equine encephalitis has been more diagnosed in the past in South America. One of the speculations is that it is misdiagnosed as dengue among other flu-like disease. The rift valley fever is more frequent in sub-Saharan Africa, and this disease has not been reported in the Americas up to the present. Whereas Nipah virus encephalomyelitis has been only reported in Asian countries and Ghana so far.

We observed that the most reported zoonoses are parasitic diseases. This might be because parasitic diseases are cheaper to diagnose compared to other diseases such as bacterial, viral, or fungal. Nevertheless, they consist of one of the more under-ascertained infections since parasites do not always cause symptoms (or alarming symptoms). We observed an increment of scientific publications from 2010, which is possibly due to the historical economic growth that Paraguay experimented the same year. This economic improvement was helped by higher exportation of agricultural products (mainly soy and beef), and likely influenced by the new presidential administration that started in 2008. After 2010, the GDP was slowly growing with some relapses in 2012, 2016 and 2019. Paraguay has had since 2014 a National Development Plan 2030 to align and achieve objectives of the "Sustainable development goals" promoted by the United Nations. Paraguay is working on three main axes supported by public politics to become a competitive and efficient country: 1) poverty reduction and social development, 2) inclusive economic growth, and 3) insertion of Paraguay in the world [103]. This plan gives Paraguayans some hope, despite the situation.

## Conclusion

In terms of DALYs, we have shown that zoonoses represent a substantial proportion of the burden of infectious diseases in Paraguay. This is further illustrated when the direct effects on animal health are also included as zDALYs. The utility of the zDALY is further supported by comparing the prioritization of diseases in terms of DALYs and zDALYs. Thus, disease prioritization varies if we consider only the human burden of diseases (DALYs) compared to the additional burden when the effects of animal diseases are also included (zDALYs). This is a clear example that we need to focus on both human and animal diseases since the zDALY provides more holistic information for disease prioritization and prevention and confirms an initial step of the "One-Health" approach to disease control. Improving data quality remains a challenge in Paraguay and crucial to approach the Paraguayan Development Plan 2030. Our results show that interdepartmental and inter-institutional communication in this country, mainly between veterinary and medical sectors, then with other related institutions such as the environment should be improved. The registration systems of diseases at national and international levels must also be improved to avoid disparities and underreporting. This country should invest in a common database for zoonoses to ease the "decision-making" and policies based on evidence in health area, including more digitalization of resources (e.g., theses) to document what is already available.

We consider that the local list of "emerging diseases" and zoonoses priorities should be updated, adding diseases such as ehrlichiosis, among other vectorial diseases.

Although DALYs and zDALYs contribute to a more holistic approach to the health losses in a population, these health metrics continue to be a challenge in terms of social and environmental impacts to make them part of an actual One Health metric.

## Supporting information

**S1 Prisma checklist.**
(DOCX)

**S1 Alternative Language Abstract.**
(DOCX)

**S1 Table. List of selected zoonoses for the systematic review.**
(XLSX)

**S2 Table. Databases and other sources searched.**
(XLSX)

**S3 Table. Monte Carlo Analysis applied to modelling of zoonotic diseases.**
(XLSX)

**S4 Table. Auxiliary information of zoonoses used to estimate the YLL, YLD and DALYs.**
(DOCX)

**S1 Text. Complete search term used according to each database.**
(DOCX)

**S1 File.** Fig A: Meta-analysis plots of zoonoses in Paraguay: forest plot and funnel plot of babesiosis in animals. Fig B: Meta-analysis plots of zoonoses in Paraguay: forest plot and funnel plot of ehrlichiosis in animals (dogs). Fig C: Meta-analysis plots of zoonoses in Paraguay: forest plot and funnel plot of leishmaniasis in animals (dogs). Fig D: Meta-analysis plots of zoonoses in Paraguay: forest plot and funnel plot of leishmaniasis in humans. Fig E: Meta-analysis plots of zoonoses in Paraguay: forest plot and funnel plot of leptospirosis in animals. Fig F: Meta-analysis plots of zoonoses in Paraguay: forest plot and funnel plot of leptospirosis in humans. Fig G: Meta-analysis plots of zoonoses in Paraguay: forest plot and funnel plot of rabies in animals. Fig H: Meta-analysis plots of zoonoses in Paraguay: forest plot and funnel plot of scabies in animals. Fig I: Meta-analysis plots of zoonoses in Paraguay: forest plot and funnel plot of toxoplasmosis in humans
(ZIP)

## Acknowledgments

Liz thanks Romina Marini, Sergio Escobar, Arami Santacruz, and Florian Graf for answering her messages and send her their work. Paraguay needs more people like you, altruistic and concerned for a better situation. Aguyje!

## Author Contributions

**Conceptualization:** Liz Paola Noguera Zayas, Simon Rüegg, Paul Torgerson.

**Data curation:** Liz Paola Noguera Zayas.

**Formal analysis:** Liz Paola Noguera Zayas.

**Funding acquisition:** Liz Paola Noguera Zayas.

**Investigation:** Liz Paola Noguera Zayas.

**Methodology:** Liz Paola Noguera Zayas, Paul Torgerson.

**Project administration:** Liz Paola Noguera Zayas.

**Resources:** Paul Torgerson.

**Supervision:** Simon Rüegg, Paul Torgerson.

**Validation:** Paul Torgerson.

**Visualization:** Liz Paola Noguera Zayas.

**Writing – original draft:** Liz Paola Noguera Zayas.

**Writing – review & editing:** Liz Paola Noguera Zayas, Simon Rüegg, Paul Torgerson.

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
