## [Decision Letter · Decision Letter 0]

2 Apr 2021

Dear Mrs Noguera Zayas,

Thank you very much for submitting your manuscript "The Burden of Zoonoses in Paraguay: A Systematic Review" for consideration at PLOS Neglected Tropical Diseases. As with all papers reviewed by the journal, your manuscript was reviewed by members of the editorial board and by several independent reviewers. In light of the reviews (below this email), we would like to invite the resubmission of a significantly-revised version that takes into account the reviewers' comments. 

We cannot make any decision about publication until we have seen the revised manuscript and your response to the reviewers' comments. Your revised manuscript is also likely to be sent to reviewers for further evaluation.

Sincerely,

Peter Horby

Associate Editor

Christine Petersen

Deputy Editor

Reviewer's Responses to Questions

**Key Review Criteria Required for Acceptance?**

**Methods**

-Are the objectives of the study clearly articulated with a clear testable hypothesis stated?

-Is the study design appropriate to address the stated objectives?

-Is the population clearly described and appropriate for the hypothesis being tested?

-Is the sample size sufficient to ensure adequate power to address the hypothesis being tested?

-Were correct statistical analysis used to support conclusions?

-Are there concerns about ethical or regulatory requirements being met?

Reviewer #1: The objectives are stated in L76-82. 

The hypothesis is weakly stated as "zoonoses represent a substantial health burden in Paraguay" but to my liking this is not testable. Reference to GDP or other cost impact would have relevance for the costs of zoonoses.

The PRISM approach describes the population of literature surveyed, and demographics for Paraguay are described. The sample size is adequate; statistical testing was not applied nor appropriate.

No ethical or regulatory concerns.

Reviewer #2: objectives are clear and the study design is in agreement with the stated objectives

**Results**

-Does the analysis presented match the analysis plan?

-Are the results clearly and completely presented?

-Are the figures (Tables, Images) of sufficient quality for clarity?

Reviewer #1: The analysis approaches the intended objective of the article (valuation of zoonoses) but there could be more made of the results. 

For example, I don’t see how the results from calculating DALYs and ALEs are integrated or at least assessed in unison. In other words, how did these results from ALEs inform our knowledge of DALYs? Surely there is more of a story to tell other than adding to the total financial impact. If the authors feel this was addressed in the article, it needs to be summarized and highlighted in the Discussion towards the end or in the Conclusions.

For the most part the results are clear and the tables and images of sufficient quality for clarity. 

However, I find that the discussion on the impact on livelihoods is thin other than to summarize that the diseases are present, recorded, and the authors can confirm that their findings reflect what is in the literature they cite (or in medical records from Paraguay). There is no wider discussion on the social burden of animal diseases in terms of direct or indirect influence on daily decision making that impacts livelihoods for example (e.g., avoidance of some species due disease risk could have economic impact on family income), or impact on family nutrition. Those are just two ideas, but those wider implications of the GBADs seem to be missing in the discussion.

I also wonder why the distributions presented in the results were not discussed in more detail. Are the authors satisfied with those distributions, do they match other findings, is there more particular work to do with regards to simulating distributions, what are the shortcomings of applying those distributions to policy work, etc. There was also no apparent effort to apply sensitivity analysis - a critical feature of robust simulation studies that is missing as far as I can tell.

Reviewer #2: results are clearley presented. 

Fig 1 map is wrong and not very useful, other countries in the region are missing, labeling is hard to read. A more informative map should be not only more complete, but should incorporate geographical data correlating to the data discussed in the text. Paraguayan regions are mentioned throughout the manuscript and it would be helpful for the reader to easily locate them on this map.

**Conclusions**

-Are the conclusions supported by the data presented?

-Are the limitations of analysis clearly described?

-Do the authors discuss how these data can be helpful to advance our understanding of the topic under study?

-Is public health relevance addressed?

Reviewer #1: See some of my previous comments.

Conclusions are very general and supported by the data but seem cursory relative to what else might be addressed.

Use of these data to advance our understanding falls short of what is promised in L79-82 although this is woven into the discussion of misdiagnosis, under estimation, under ascertainment and (this was not clearly noted but was suggested) access to health care for marginalized and remote communities. Those are all good points that need to be brought to conclusion - can these results influence policy in some way to correct those problems?

Public health relevance is addressed throughout the paper.

Reviewer #2: (No Response)

**Editorial and Data Presentation Modifications?**

Reviewer #1: Grammar and punctuation: 

There are many small punctuation and occasional grammatical errors that should be addressed by a fluent speaker/writer of English. For example, line 11 is not a proper sentence, “One Health” is not hyphenated and should be capitalized, numbers greater than 100 should use commas, “31 thousand” etc. should be written as 31,000 (million is acceptable as a word), “/year” in the text should be spelled out as “per year”, disease names are not capitalized unless proper nouns are used, “decision making” is not hyphenated, “12 %” should have no space as in “12%”, “potential loses” should read “potential losses”, “and in less quantity sheep” is awkward and simply not stated, line 68 probably refers to livestock product exports but needs to state so, conjunctions and prepositions are missing or incorrect throughout the document, etc. There are far too many edits to list them all – please have someone review and correct these errors.

Reviewer #2: (No Response)

**Summary and General Comments**

Reviewer #1: This research has promise to present more robust and extended findings if my comments are taken into consideration. As it stands is presents more as a catalogue of estimations with reflection on which diseases are already noted as zoonotic and a burden in Paraguay. That seems to be the overall weakness of the paper.

The strength is the wide search for data in the literature and an effort to model disease distributions and impact using simulation.

L57: I don’t think Fig 1 is particularly helpful considering how much page space this colour illustration occupies. Readers will either know where Paraguay is located, or can find this quickly on the internet. There is nothing to indicate livestock density on this map or relative economic value compared to neighbours, which might have made it more interesting.

L85: A few of the zoonoses chosen could be more narrowly defined (E. coli, tuberculosis, and avian influenza for example). You do note that for some diseases this is not available but surely you there is some reference to these distinctions in the literature. For a paper that addresses zoonoses, this seems a critical point of interest. Tuberculosis from M. bovis vs. M. tuberculosis, or HPAI vs. non-pathogenic AI will change weighting factors and output measurements. This should be addressed in general somewhere in your article (cf. my comment L148).

L98: Your PRISMA approach is a well accepted approach and you have conducted what appears to be an extensive search. However, with so many diseases listed compared to the number of articles, I wonder if you might consider a figure that shows clusters of disease over time (e.g., bacterial, viral, common, rare, self-limiting, fatal without intervention, etc.)

L133, L146: Equations would help here. These are not complicated concepts but the explanations are hard to follow for someone not familiar with the equations.

L148: Further to my comment L85, do you not have this distinction for Paraguay? It would be good to know if that is the case. Using all of the Americas as a reference is a pretty wide reference base.

L170: I am not sure from reference where your pet values came from – no mention pet market sources or of social media that I can see in your S2 info. (The livestock sources seem reliable and reasonable.)

L172: With respect to simulation, your terminology gets a bit sloppy. Technically, uncertainty cannot be estimated because you do not know either probability or outcome, but you could estimate risk (the product of probability and outcome). What are you estimating? Risk of …? This needs to be stated clearly in the first sentence of the paragraph starting at L172.

L172: Why did you decide on beta, gamma, etc. distributions? Expert opinion, best fit (and then how and of what other distributions), past experience? Why were some diseases modeled under two or more distributions and others only one? A lot of assumptions seem to be missing here. Seed, underlying estimator (MLE, etc.), etc.

L174: I believe what you state as “draws” should be iterations.

L182: Is this in reference to data you refer to previously or is this a new reference? Please clarify and if new, information regarding those sources is needed.

L184: Why were these particular diseases selected?

L185: Is the “random effect model” approach something all readers should know? Some clarity is needed here.

L195: Unless you tested for significance, replace that word with “substantial”. Is there a reason to account for this substantial increase?

L234: … as well as differentiation between bacterial species and their relative impact on human vs. animal populations. It seems to me this is the Achilles heel of the research – interesting from a global conclusion but weak specificity of local conclusions (to use the modeling reference to global vs. local optima points).

L241: Can this not be incorporated into your estimation of the impact of “tuberculosis” somehow, or perhaps model both TB and bTB? I see you have approached that notion in L337-338.

L249-252: I don’t see the point of this paragraph if these individuals emigrated (out of Paraguay). Please add clarification of relevance.

L373: Agreed, but neither is the emotional value of human life incorporated into DALYs. As for wild animals, you could have estimated marketable valuation or value based on economic substitution.

L392-400: A substantial burden from rabies is the cost to the public sector or PEP treatment as well as dog capture, neuter, vaccinate, and release programs. Lots of articles in the rabies literature to draw on from S. America that you could use just to note that is a cost not incorporated in your study. You might start with: de Carval et al. (2018) Rabies in the Americas: 1998-2014. PNTD. https://doi.org/10.1371/journal.pntd.0006271. Chile is now considered free of dog rabies, but it took a strong and costly campaign to get there – should be data on that too.

Reviewer #2: In this paper, the authors assess the burden of zoonoses in Paraguay. They address the influence of zoonoses on both human and animal health and its socioeconomic impact, also they discuss its national/international importance in terms of economic interest and healthcare priorities.

A systematic review is performed based on the previously reported systematic review and meta analyses: PRISMA. Obtained data was used to estimate DALY and zDALY, useful parameters to assess the burden of zoonoses in Paraguay.

I would like to discuss some points that were not clear for me: In cases of comorbidity due to more than one infection, how was the employed criteria?

I agree that “The five most important pathogens of DALYs in decreasing order constitutes E. coli, Trypanosoma cruzi, Leishmania spp, Toxoplasma gondii and Campylobacter spp.. They are responsible for 75% of the disease burden.” However, E. coli and T. cruzi are by far, the most important pathogens, they should be discussed separately since these two pathogens represent a half of the total disease burden. 

Should be considered T. cruzi to discuss zDALY? As far as I know, Chagas disease doesn’t affect cattle, poultry, pigs, sheep, horses, goats or any other economically important animal. 

More detail is necessary to establish Paraguay as a unique case for this study. When appropriate, available information from other countries should be included for comparison. If it is the purpose of the authors, as stated, to uncover clues in order to set disease control priorities, cross-country comparisons become even more essential. What is the burden of the analyzed zoonoses in neighboring countries? What is the disease burden in non-neighboring countries which share the same type of economy heavily skewed towards agricultural production?

Minor comments:

Large sections (particularly in the introduction) should be revised for grammar and syntax. Oxford commas are inconsistently used throughout.

Line 11: Change “this difficult to gauge”… by “it is difficult to gauge…”

Line 65: losses

Line 166: Was the economic value of cattle estimated in export prices or prices for the local market, or a combination of both?

Line 167: “In respect of pets” change by “with respect to pets” 

257. Replace “have been interviewed” with “were interviewed”

262. It would be useful to mention the legal status of disclosure of risks to workers in the country in this context.

Line 285: lose

Line 383: Discussion of animal bites seems out of place outside of the context of rabies.

PLOS authors have the option to publish the peer review history of their article (what does this mean?). If published, this will include your full peer review and any attached files.

Reviewer #1: No

Reviewer #2: Yes: Leticia Perez-Diaz
---

## [Decision Letter · Decision Letter 1]

3 Sep 2021

Dear Mrs Noguera Zayas,

Thank you very much for submitting your manuscript "The Burden of Zoonoses in Paraguay: A Systematic Review" for consideration at PLOS Neglected Tropical Diseases. As with all papers reviewed by the journal, your manuscript was reviewed by members of the editorial board and by several independent reviewers. The reviewers appreciated the attention to an important topic. Based on the reviews, we are likely to accept this manuscript for publication, providing that you modify the manuscript according to the review recommendations. 

Sincerely,

Peter Horby

Associate Editor

Christine Petersen

Deputy Editor

Reviewer's Responses to Questions

**Key Review Criteria Required for Acceptance?**

**Methods**

-Are the objectives of the study clearly articulated with a clear testable hypothesis stated?

-Is the study design appropriate to address the stated objectives?

-Is the population clearly described and appropriate for the hypothesis being tested?

-Is the sample size sufficient to ensure adequate power to address the hypothesis being tested?

-Were correct statistical analysis used to support conclusions?

-Are there concerns about ethical or regulatory requirements being met?

Reviewer #1: I am finding it difficult to locate the changes, but I think I succeeded. The version I was sent has no changes indicated in any way. Nor do the authors indicate on what line(s) changes were incorporated in the new manuscript. This would have been helpful.

I will refer to my previous comments and author replies.

The authors' response focusing on GDP misses my earlier point - I did not suggest you measure GDP (which, by the way, is an economic metric, not monetary metric). I suggest you need to incorporate some sort of standardized cost measure which would be consistent with your frequent reference to the economic value and market valuation of animals and their use. This to me is one of the weaknesses in general of adopting DALYs or modified DALYs - animals are property with market value but people are not.

This boils down to a philosophical and intellectual debate regarding how to measure and incorporate market valuation. DALYs and your zDALY certainly help, but the wider social impact of the market losses is not captured (e.g., food security and all the other SDGs). I suggest you simply add a line that notes that somewhere in your paper (end of Discussion or Conclusions perhaps). No one has achieved that in a single index and that is your challenge.

Reviewer #2: objectives are clear and the study design is in agreement with the stated objectives. The population is adequate to test the hypothesis.

In the new version submitted, the methodology employed is better explained which makes it clearer.

**Results**

-Does the analysis presented match the analysis plan?

-Are the results clearly and completely presented?

-Are the figures (Tables, Images) of sufficient quality for clarity?

Reviewer #1: Again, it's quite difficult to find the specific changes without either TrackChanges or author reference to the line(s) where changes were made. In future, I suggest the lead author indicate to reviewers what lines in the new manuscript incorporate changes.

I cannot see where you discuss the social burden of disease in your discussion or conclusion. It may well be beyond the scope of your study, but I do think it bears mentioning as a weakness of current methods of estimation of GBADs. If not already incorporated, please add to the Discussion in a clearly stated sentence. There are many such methods used in development/ growth/ welfare economics, but just mentioning that weakness (absence) I think is important.

Reviewer #2: to clarify, new graph and figure were added to show clusters of disease (publications) from 2000 to 2020 and the relative impact of diseases in the country (DALY, ALE and zDALY)

**Conclusions**

-Are the conclusions supported by the data presented?

-Are the limitations of analysis clearly described?

-Do the authors discuss how these data can be helpful to advance our understanding of the topic under study?

-Is public health relevance addressed?

Reviewer #1: I think your Conclusions is better stated in your revised manuscript.

Reviewer #2: Conclusions were rewritten and looks more clear. They are supported by presented data

**Editorial and Data Presentation Modifications?**

Reviewer #1: (No Response)

Reviewer #2: (No Response)

**Summary and General Comments**

Reviewer #1: Other changes to the manuscript (e.g., added graph, equations, etc.) have improved the manuscript.

Reviewer #2: In this paper, the authors quantify the impact of different pathogens in human and animal disease burden in Paraguay, through the estimation of DALY and DALYz parameters respectively.

The methodology is now better explained and clearer in this new version of the manuscript.

The Discussion Section has been also improved. The authors well discuss the limitations of the methods and the different possible causes of underestimation or overestimation of the presented data. Moreover, each one of the pathogens causing the studied zoonoses are now described in the discussion. In this context, I think that the discussion is now quite long (too much). The authors should focus on the description of the most relevant zoonoses (the high priority) in the study, those with higher DALY and DALYz values. 

The study of the burden of zoonosis diseases and the interconnection between people and animals will help to develop optimal health outcomes, prioritizing those illness with higher impact in human and animal health minimizing economic losses. For that reason I think that this work is very interesting to be published but I strongly suggest shortening the section Discussion to accept the manuscript.

minor comments:

E coli must be defined by Escherichia coli by the first time

Lane 234. From results in Table 1, one the most relevant pathogens, considering the zDALY calculation was T. cruzi, however, there is no ALE data in the table (from dogs) so in this case, zDALY is the same as DALY. Is zDALY discussion relevant in this case?

In figure 3a, 3b and 3c, What is the meaning of the numbers in the right upper corner? are they percentages of the estimated total DALY, Ale and zDALY respectively? Please, explain

PLOS authors have the option to publish the peer review history of their article (what does this mean?). If published, this will include your full peer review and any attached files.

Reviewer #1: Yes: David C. Hall

Reviewer #2: Yes: Leticia Pérez-Díaz

Figure Files:

Data Requirements:

Reproducibility:

References

---

## [Decision Letter · Decision Letter 2]

15 Oct 2021

Dear Mrs Noguera Zayas,

We are pleased to inform you that your manuscript 'The Burden of Zoonoses in Paraguay: A Systematic Review' has been provisionally accepted for publication in PLOS Neglected Tropical Diseases.

Best regards,

Christine A Petersen

Deputy Editor

Christine Petersen

Deputy Editor

Reviewer's Responses to Questions

**Key Review Criteria Required for Acceptance?**

**Methods**

-Are the objectives of the study clearly articulated with a clear testable hypothesis stated?

-Is the study design appropriate to address the stated objectives?

-Is the population clearly described and appropriate for the hypothesis being tested?

-Is the sample size sufficient to ensure adequate power to address the hypothesis being tested?

-Were correct statistical analysis used to support conclusions?

-Are there concerns about ethical or regulatory requirements being met?

Reviewer #1: (No Response)

**Results**

-Does the analysis presented match the analysis plan?

-Are the results clearly and completely presented?

-Are the figures (Tables, Images) of sufficient quality for clarity?

Reviewer #1: (No Response)

**Conclusions**

-Are the conclusions supported by the data presented?

-Are the limitations of analysis clearly described?

-Do the authors discuss how these data can be helpful to advance our understanding of the topic under study?

-Is public health relevance addressed?

Reviewer #1: (No Response)

**Editorial and Data Presentation Modifications?**

Reviewer #1: (No Response)

**Summary and General Comments**

Reviewer #1: (No Response)

PLOS authors have the option to publish the peer review history of their article (what does this mean?). If published, this will include your full peer review and any attached files.

Reviewer #1: No

---

## [Editor Report · Acceptance letter]

28 Oct 2021

Dear Mrs Noguera Zayas,

We are delighted to inform you that your manuscript, "The Burden of Zoonoses in Paraguay: A Systematic Review," has been formally accepted for publication in PLOS Neglected Tropical Diseases.

Best regards,

Shaden Kamhawi

co-Editor-in-Chief

Paul Brindley

co-Editor-in-Chief
